# Evaporation of microwave-shielded polar molecules to quantum degeneracy

Andreas Schindewolf[1,2], Roman Bause[1,2], Xing-Yan Chen[1,2], Marcel Duda[1,2], Tijs Karman[3], Immanuel Bloch[1,2,4] & Xin-Yu Luo[1,2 ✉]

Ultracold polar molecules offer strong electric dipole moments and rich internal structure, which makes them ideal building blocks to explore exotic quantum matter[1–9], implement quantum information schemes[10–12] and test the fundamental symmetries of nature[13]. Realizing their full potential requires cooling interacting molecular gases deeply into the quantum-degenerate regime. However, the intrinsically unstable collisions between molecules at short range have so far prevented direct cooling through elastic collisions to quantum degeneracy in three dimensions. Here we demonstrate evaporative cooling of a three-dimensional gas of fermionic sodium–potassium molecules to well below the Fermi temperature using microwave shielding. The molecules are protected from reaching short range with a repulsive barrier engineered by coupling rotational states with a blue-detuned circularly polarized microwave. The microwave dressing induces strong tunable dipolar interactions between the molecules, leading to high elastic collision rates that can exceed the inelastic ones by at least a factor of 460. This large elastic-to-inelastic collision ratio allows us to cool the molecular gas to 21 nanokelvin, corresponding to 0.36 times the Fermi temperature. Such cold and dense samples of polar molecules open the path to the exploration of many-body phenomena with strong dipolar interactions.

The field of ultracold polar molecules is on the verge of entering an exciting phase[14,15]. The possibilities of quantum simulation that come within experimental reach range from $p$-wave superfluids[1–4], super-solids[5,6] and Wigner crystals[7,8], to novel spin systems and extended Hubbard models[9]. Many of these proposals require a deeply degenerate quantum gas of molecules at tens of nanokelvin with strong dipolar interactions in three dimensions (3D).

Although non-interacting quantum-degenerate gases of polar molecules have been produced by assembling degenerate atomic mixtures[16–18], active cooling to the quantum-degenerate regime has remained challenging. The main hurdle towards efficient evaporative cooling of ultracold molecules is posed by their rapid collisional loss. In particular, when the molecules are polarized by an electric field in three-dimensional traps they tend to collapse in attractive head-to-tail collisions[19]. It turns out that even molecules that are nominally stable against chemical reactions[20] undergo inelastic collisions when they reach short range. The exact nature of these loss processes is still not fully understood and remains under investigation[21–26].

Collisional loss in short-range encounters between molecules can be suppressed by engineering repulsive interactions using external fields[27–32]. Recently, a molecular gas of $^{40}K^{87}Rb$ was stabilized and evaporatively cooled to below the Fermi temperature by applying a strong d.c. electric field and confining the motion of the molecules to two dimensions (2D), which effectively prevented attractive collisions along the third dimension[33]. For molecules in 3D, reduced collisional loss has been demonstrated by using a specific d.c. electric field[34] to bring rotational states of colliding molecules in resonance with each other. However, attempts to produce a degenerate quantum gas of polar molecules through collisional cooling has so far remained unsuccessful in 3D owing to a low elastic-to-inelastic collision ratio as well as a relatively low initial phase-space density[34,35]. Repulsive barriers between molecules can also be engineered by applying blue-detuned circularly polarized microwaves[36,37], which has been successfully used recently to reduce collisional loss between two calcium monofluoride molecules in an optical tweezer[38].

In this work, we induce fast elastic dipolar collisions in a quantum gas of fermionic $^{23}Na^{40}K$ molecules dressed by a circularly polarized microwave field while strongly suppressing inelastic collisions in all three dimensions. The elastic collision rate increases dramatically with the effective lab-frame dipole moment, which can be adjusted with the microwave power and detuning. Under appropriate conditions, we find that the elastic collision rate can be about 500 times larger than the inelastic collision rate, and even exceed the trap frequencies such that the molecular gas enters the hydrodynamic regime. We evaporate a three-dimensional near-degenerate sample of molecules to 21(5) nK, which corresponds to 36(9)% of the Fermi temperature $T_F$, deep in the quantum-degenerate regime. At such low temperatures, the inelastic collision rate between molecules becomes negligible, leading to a lifetime of the degenerate molecular sample of up to 0.6 s.

[1]Max-Planck-Institut für Quantenoptik, Garching, Germany. [2]Munich Center for Quantum Science and Technology, Munich, Germany. [3]Institute for Molecules and Materials, Radboud University, Nijmegen, The Netherlands. [4]Fakultät für Physik, Ludwig-Maximilians-Universität, Munich, Germany. ✉e-mail: xinyu.luo@mpq.mpg.de

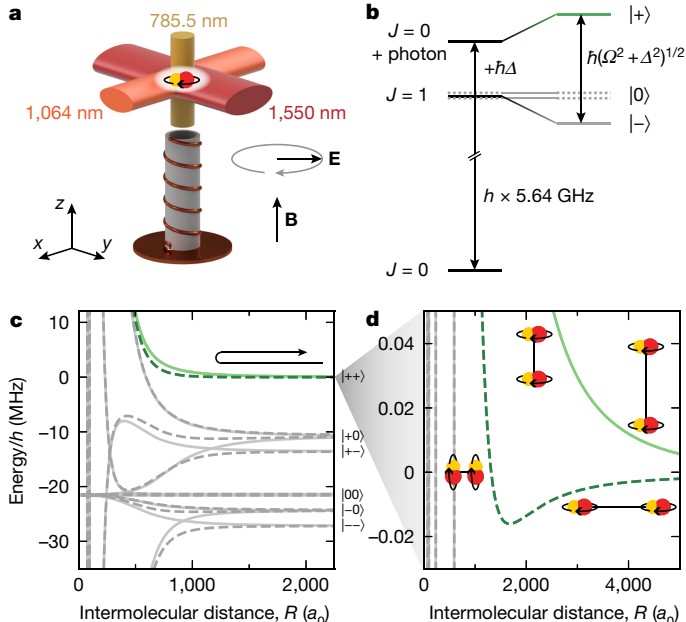

**Fig. 1 | Microwave shielding. a**, Sketch of the experimental set-up. The molecules are confined by up to three optical dipole traps with the indicated wavelengths. A helical antenna emits the rotating electric field **E**. A d.c. magnetic offset field **B** is generated by a pair of coils (not shown). **b**, Preparation of the molecules in the dressed state $|+\rangle$. The microwave detuning $\Delta$ is typically comparable to the Rabi frequency $\Omega$. **c**, Interaction potentials for $\Delta = 2\pi \times 8$ MHz and $\Omega = 2\pi \times 11$ MHz. The solid and dashed lines show the potential energy of molecules colliding along ($\theta = 0$) and orthogonal ($\theta = 90°$) to the microwave propagation direction, respectively. The centrifugal barrier is omitted for clarification. The arrow illustrates an elastic collision. Here, $a_0$ is the Bohr radius. **d**, Close-up of the shielded collisional channel $|++\rangle$. The insets show the alignment of the rotating molecules with respect to each other. At intermediate range, the molecules align with respect to the intermolecular axis.

## Microwave shielding and dipolar elastic collisions

To realize shielding in 3D, the two lowest rotational states of the molecules can be coupled via a blue-detuned circularly polarized microwave[36]. The dressed state is a time-dependent superposition of rotational states that features an induced rotating dipole moment, which follows the strong a.c. electric field of the microwave. Molecules prepared in this state exhibit an effective dipolar interaction at long range. When they approach each other, the direct coupling between the induced dipoles starts to dominate, forming a repulsive barrier that effectively shields the molecules from detrimental inelastic collisions at short range.

In our experiment, the microwave field is generated by a helical antenna, as illustrated in Fig. 1a. The antenna emits a mainly $\sigma^-$-polarized microwave that couples the rotational ground state $\left| J=0, m_J=0, \delta N_p=0 \right\rangle$ to the excited state $|1, -1, -1\rangle$, as shown in Fig. 1b. Here, $J$ is the rotational quantum number, $m_J$ is its projection on the magnetic field axis and $\delta N_p$ is the change in the number of photons in the microwave field. The frequency of the microwave is approximately given by the rotational constant $B_{rot} = h \times 2.822$ GHz (ref. [39]) and the detuning from resonance $\Delta$ as $2B_{rot}/h + \Delta/(2\pi)$, where $h$ is Planck's constant. Coupling the rotational states creates the dressed states[40]

$$\begin{aligned} |+\rangle &= \cos\varphi|0,0,0\rangle + \sin\varphi|1,-1,-1\rangle, \\ |-\rangle &= -\sin\varphi|0,0,0\rangle + \cos\varphi|1,-1,-1\rangle, \end{aligned} \quad (1)$$

with the mixing angle $\varphi = \arctan((\Delta + \sqrt{\Delta^2 + \Omega^2})/\Omega)$ where $\Omega$ is the Rabi frequency. Rotationally excited states with $m_J = 0$ and $m_J = 1$, to which the microwave does not couple, remain as spectator states $|0\rangle$.

On resonance, that is, at $\Delta = 0$, the coupling strength $\hbar\Omega \approx h \times 11$ MHz (where $\hbar$ is the reduced Planck's constant) defines the splitting of $|+\rangle$ and $|-\rangle$. We typically choose a blue detuning $\Delta$ on the order of $\Omega$ to prepare the molecules in the $|+\rangle$ state (for details on the microwave transition, the coupling strength and the dressed-state preparation, see Methods and Supplementary Information).

Molecules that interact in the upper dressed state through their induced rotating dipoles are described by the collisional channel $|++\rangle$, shown in Fig. 1c,d. At long range, their time-averaged interaction energy is given by[40]

$$V_{dd} = -\frac{d_0^2}{4\pi\varepsilon_0} \frac{1-3\cos^2\theta}{12(1+(\Delta/\Omega)^2)R^3}, \quad (2)$$

where $d_0 \approx 2.7$ Debye is the intrinsic dipole moment of NaK, $\varepsilon_0$ is the vacuum permittivity, $R$ is the intermolecular distance and $\theta$ denotes the angle between the rotation axis of the induced dipole and the intermolecular axis. Molecules thus acquire an effective dipole moment $d_{eff} = d_0/\sqrt{12(1+(\Delta/\Omega)^2)}$. It is noted that the sign of the interaction energy is inverted compared with the interaction between two regular dipoles with dipole moment $d_{eff}$.

At intermediate range, the dipole–dipole interaction dominates, so that the molecules orient themselves with respect to the intermolecular axis, rather than along the rotating electric field[41]. The reorientation is made possible by contributions of the spectator states $|0\rangle$. As the molecules are prepared in the upper dressed state, they couple at this point to the repulsive branch of the dipole–dipole interaction, which shields the molecules from reaching short range regardless of their initial angle of approach[42]. In fact, the remaining two-body loss is dominated by non-adiabatic transitions to lower-lying field-dressed states, such as $|+0\rangle$, which is accompanied by an energy release on the order of $\hbar\Omega$ and suffices to eject molecules from the optical dipole trap.

The performance of evaporative cooling is ultimately limited by the ratio $\gamma$ of elastic-to-inelastic two-body collision rates. We characterize the inelastic collision rate coefficient $\beta_{in}$ by measuring the two-body decay of the molecules in a thermal gas at temperatures $T$ around 800 nK. The molecules are initially formed from ultracold atoms by means of a magnetic Feshbach resonance and subsequent stimulated Raman adiabatic passage (STIRAP) to their absolute ground state[43] (see Methods for more details on the experimental conditions and the preparation of the molecular samples). For most measurements, the initial average density $n_0$ is about $3.0 \times 10^{11}$ cm$^{-3}$. The two-body loss is relatively high for hot and dense molecules, which helps us to distinguish it from one-body loss. The latter is mainly caused by the coupling to other dressed states owing to the phase noise of the microwave, which results in an exponential decay with a time constant of about 600 ms when the two-body loss is small (Methods). We determine the elastic collision rate coefficient $\beta_{el}$ by applying parametric heating to the molecular sample along the vertical direction and measuring the cross-dimensional thermalization of effective temperatures $T_h$ and $T_v$, as shown in Fig. 2b (see also Methods). $T_h$ and $T_v$ are defined along the horizontal ($x$ and $y$) and the vertical ($z$) directions, respectively. The rate coefficients $\beta_{el}$ and $\beta_{in}$ are extracted from the time evolution of the measured molecule number $N$ and of the temperatures $T_h$ and $T_v$ by fitting a set of coupled differential equations that model the molecule losses and the cross-dimensional rethermalization (Methods). The results of the measurements are compared with coupled-channel calculations for a thermal sample at $T = 800$ nK and for a degenerate sample at $30$ nK $= 0.4T_F$, as shown in Fig. 2. The calculations account for a residual ellipticity of the microwave polarization (Supplementary Information).

In absence of the microwave field, we find $\beta_{in} = 7.7(5) \times 10^{-11}$ cm$^3$ s$^{-1}$ for $T = 800$ nK, which is in reasonable agreement with the calculated value of $4.9 \times 10^{-11}$ cm$^3$ s$^{-1}$. The shielding is most efficient at $\Delta = 2\pi \times 8$ MHz,

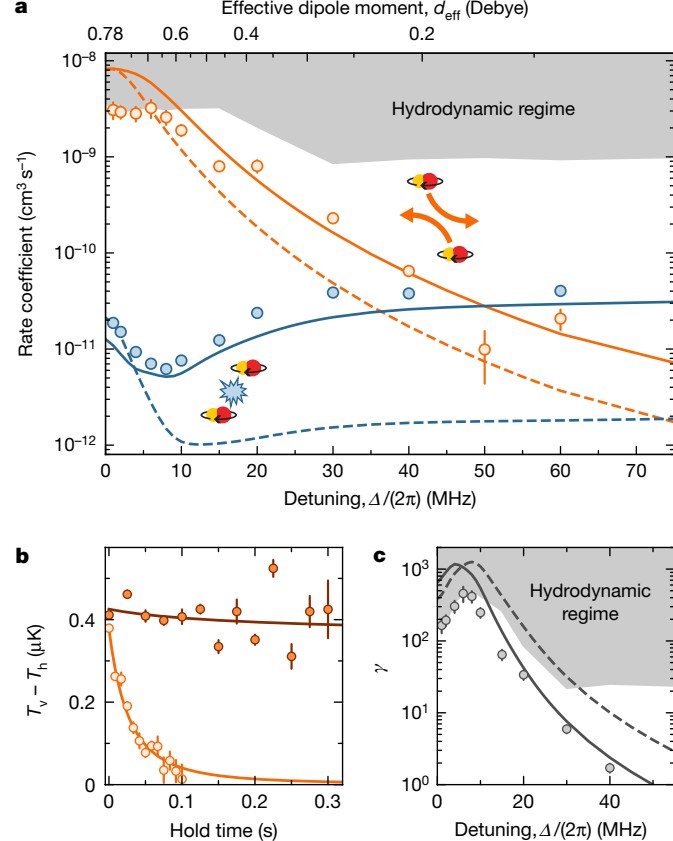

**a**

Effective dipole moment, $d_{eff}$ (Debye)

Rate coefficient ($cm^3 s^{-1}$)

Detuning, $\Delta/(2\pi)$ (MHz)

Hydrodynamic regime

**b**

$T_v - T_h$ ($\mu K$)

Hold time (s)

**c**

$\gamma$

Detuning, $\Delta/(2\pi)$ (MHz)

Hydrodynamic regime

**Fig. 2 | Elastic and inelastic collisions. a**, Rate coefficients for elastic (orange) and inelastic (blue) scattering events. The solid and dashed lines show coupled-channel calculations for a thermal sample with $T$ = 800 nK and a degenerate sample with $T$ = 30 K = $0.4T_F$, respectively. The data points show the measurement results for $T$ = 800 nK. The shaded area indicates the limit for measurements of the elastic collision rate imposed by the hydrodynamic regime. On resonance, the Rabi frequency is $\Omega \approx 2\pi \times 11$ MHz. The triangular markers on the side indicate the calculated inelastic rate coefficients in the absence of a microwave field. The error bars are the standard deviation from the fit to the differential equations. **b**, Examples of cross-thermalization measurements at $\Delta = 2\pi \times 30$ MHz (bright) and $\Delta = 2\pi \times 80$ MHz (dark). In the latter case, $\beta_{el}$ is consistent with zero within the error bars. The error bars are the standard error of the mean of four repetitions. The lines are fits of a coupled differential-equation system modelling the collision rates (Methods). **c**, The ratio $\gamma$ of elastic-to-inelastic collision rates based on the measurements and calculations presented in **a**. The error bars denote the standard deviation.

where $\beta_{in}$ drops to $6.2(4) \times 10^{-12}$ cm$^3$ s$^{-1}$, corresponding to an order of magnitude suppression of the two-body losses, as illustrated in Fig. 2a.

For spin-polarized fermionic polar molecules, the elastic collision rate is dominated by dipolar scattering. The corresponding scattering cross-section scales approximately as $d_{eff}^4$ (Supplementary Information) and can therefore be tuned over multiple orders of magnitude with the detuning of the microwave. In the regime of weak interactions, that is, at large detunings, the rethermalization rate is proportional to $\beta_{el}$. However, for $\Delta \lesssim 2\pi \times 10$ MHz, the elastic collisions become so frequent that the mean free path of the molecules is less than the size of the molecular cloud, even though here we intentionally reduced the initial density to $n_0 = 0.7 \times 10^{11}$ cm$^{-3}$. In this hydrodynamic regime, the rethermalization rate is limited to about $\bar{\omega}/(2\pi) \approx 120$ Hz, where $\bar{\omega}$ is the geometric mean trap frequency[44]. Consequently, measured values of $\beta_{el}$ saturate near the so-called hydrodynamic limit $N_{col}\bar{\omega}/(2\pi n_0)$ where $N_{col} \approx 2$ is the average number of collisions required for rethermalization in our system (Methods). This also limits the maximum measured

value of $\gamma$ to 460(110), as illustrated in Fig. 2c. Away from the hydrodynamic regime, we find excellent agreement between the experimentally determined and the calculated values of $\beta_{el}$ and $\gamma$ for $T$ = 800 nK. Our calculations show that $\gamma$ can exceed 1,000 for ideal values of $\Delta$. In the future, it should be possible to improve the shielding and achieve $\gamma \approx 5,000$ by optimizing the purity of the microwave polarization (Supplementary Information).

## Evaporative cooling to deep quantum degeneracy

With $\gamma \gtrsim 500$ at the optimum shielding detuning $\Delta = 2\pi \times 8$ MHz, the evaporative cooling of our molecular sample is straightforward. We start with a low-entropy but non-thermalized sample of about $2.5 \times 10^4$ molecules produced from a density-matched degenerate atomic mixture[18]. The power of the two horizontally propagating laser beams, which hold the molecules against gravity and thereby define the effective trap depth $U_{trap}$, is lowered exponentially over the course of 150 ms. The hottest molecules can then escape the dipole trap in the direction of gravity. The remaining molecules rethermalize through elastic dipolar collisions, effectively reducing $T$ and $T/T_F$. During evaporation, the calculated elastic collision rate is about 500 Hz, which is much higher than the trap frequencies. Therefore, the rethermalization rate saturates to around $\bar{\omega}/(2\pi) \approx 60$ Hz. To maintain a high rethermalization rate while lowering the trap depth, we reinforce the horizontal confinement by exponentially ramping up the power of an additional beam propagating along the vertical direction.

We characterize the evaporation by varying the final trap depth. Figure 3a shows $T$ and $T/T_F$ against the number of remaining molecules $N$ after 150 ms of forced evaporation. The values of $T$ and $T/T_F$ are deduced from a polylogarithmic fit to the momentum distribution of the sample, which is imaged after 10 ms time of flight (Methods). If the trap depth is not reduced, the interacting molecules will thermalize but are not forced to evaporate. Initially the molecules exhibit a sloshing motion in the trap due to photon-recoil transfer from the STIRAP pulses. Damping of such collective excitations and particle loss substantially reduce the phase-space density of the sample. After 150 ms holding at the initial trap depth, we obtain $1.43(5) \times 10^4$ molecules at a temperature of 176(5) nK with $T/T_F = 1.00(3)$ (Fig. 3d). If we instead evaporate to $3.6(3) \times 10^3$ molecules by reducing the final trap depth to about $k_B \times 250$ nK, where $k_B$ is the Boltzmann constant, we reach a temperature of 38(2) nK with $T/T_F = 0.47(2)$ (Fig. 3e). However, the evaporation has not stopped at this point. We can hold the molecular sample for an additional hold time $t_h$ in the trap to make use of plain evaporation, as shown in Fig. 3b,c. The plain evaporation lasts for about 100 ms. Thereafter, the molecule number exhibits an exponential decay with a 1/e lifetime of about 600 ms. At $t_h$ = 150 ms, we measure a temperature of 21(5) nK with $T/T_F = 0.36(9)$ (Fig. 3f).

We checked that the temperatures obtained from the Fermi–Dirac fit are consistent with those deduced from a Gaussian fit to the thermal wing, in which a possible influence of the dipolar interactions is relatively small (Methods). Interestingly, the optical density in a small region of the cloud centre is higher than the Fermi–Dirac fit in the coldest sample III. However, the low signal-to-noise ratio does not allow us to conclusively establish whether this is owing to imaging noise or potential dipolar interaction effects. In the future, it will be interesting to investigate the underlying mechanism.

## Discussion

As a result of the large effective dipole moment of the microwave-dressed NaK molecules, our highest measured value of $\gamma$ of about 500 is a factor of 40 larger than the ratio realized in previous experiments in 3D[34] and almost twice as large compared with experiments in 2D[33]. Such optimal collisional parameters facilitate the efficient evaporation of NaK molecules to below $0.4T_F$.

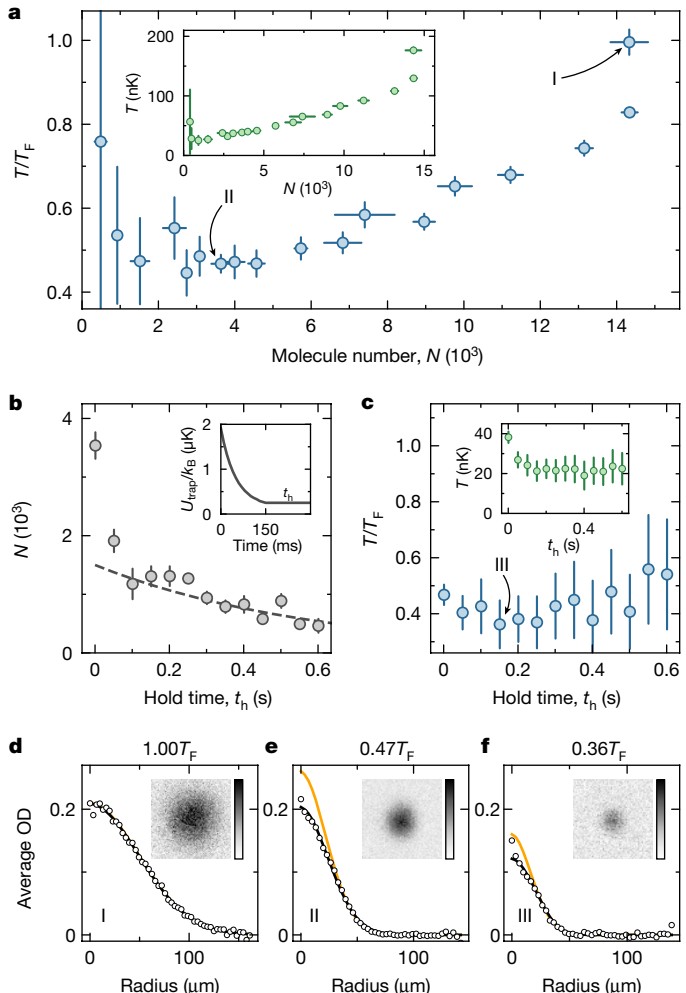

**Fig. 3 | Evaporation. a**, $T/T_F$ and $T$ (inset) against the remaining number of molecules $N$ after 150 ms of evaporative cooling for various final trap depths. **b**, Plain evaporation during the hold time $t_h$ after 150 ms of forced evaporation. The dashed line indicates the one-body decay and the inset sketches the evolution of the effective trap depth $U_{trap}$. **c**, $T/T_F$ and $T$ (inset) during the hold time $t_h$. The error bars of $N$ in **a** and **b** are the standard error of the mean of 5–20 repetitions. The error bars of $T$ in **a** and **c** are the standard deviation from the fit to the averaged images. **d**–**f**, Azimuthally averaged optical density (OD) of the samples after 10 ms time of flight. The samples are prepared under the evaporation conditions I (**d**), II (**e**) and III (**f**). The images (insets) of the samples are averages from 5 (**d**), 20 (**e**) or 6 (**f**) individual images. The black lines show polylogarithmic fit functions, and the orange lines are fits of a Gaussian to the thermal wings of the sample.

With the coldest samples realized in our experiment, the dipolar interaction in the system corresponds to about 5% of the Fermi energy. This is three times higher than what was reached in degenerate Fermi gases of magnetic atoms[45]. In the near future, intriguing dipolar many-body phenomena such as modifications of collective excitation modes[46], distortion[45] or the collapse[47] of the Fermi sea should be observable in suitable trap geometries and with improved detection of the cloud expansion.

In ref. [18], based on particle loss, we estimated the initial temperature of the non-thermalized ground-state molecules to be $0.52T_F$. However, damping of collective excitations during the rethermalization in combination with more particle loss leads to a sample temperature of about $1T_F$ after 150 ms if we do not force evaporation. The performance of the evaporation can consequently be further improved by implementing the following measures: first, the initial phase-space density can be

increased by optimizing the STIRAP transfer. Second, lower phase noise and better polarization purity of the microwave field should lead to reduced one- and two-body losses during evaporation. Finally, new strategies of evaporation in the hydrodynamic regime need to be explored to accelerate the thermalization process[44]. With these upgrades, it should be possible to reach temperatures below $0.1T_F$, where many intriguing quantum phases are expected[1–9]. In particular, fermionic polar molecules can pair up and form a superfluid with an anisotropic order parameter and even a Bose–Einstein condensate of tetramers[1,2,4]. Up to now, such scenarios have rarely been theoretically investigated because it was believed that polar molecules could not be sufficiently stable under conditions where both attractive and repulsive interactions play an important role.

## Conclusion

We have demonstrated a general and efficient approach to evaporatively cool ultracold polar molecules to deep quantum degeneracy in 3D by dressing the molecules with a blue-detuned circularly polarized microwave, achieving very low temperatures together with strong tunable dipolar interactions. The simplicity of the technical set-up makes our method directly applicable in a wide range of ultracold-molecule experiments. Our results point to an exciting future of long-lived degenerate polar molecules for investigating quantum many-body phases with long-range anisotropic interactions and for other applications in quantum sciences.

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

## Methods

### Sample preparation

To create our molecular samples, we first prepared a density-matched double-degenerate mixture of $^{23}$Na and $^{40}$K atoms. The atoms were subsequently associated to weakly bound molecules by means of a magnetic Feshbach resonance. Finally, the molecules were transferred to their absolute ground state via STIRAP. Details about the preparation process are described in refs. [18,43]. At the beginning of the measurements described in the main text, the molecules were trapped by the 1,064-nm and the 1,550-nm beam shown in Fig. 1a at a d.c. magnetic field of 72.35 G.

For the measurements of the collision rates, the microwave transition strength and to characterize the one-body loss, we worked with thermal molecules and sometimes reduced the molecule number to suppress interactions. For the collision rate measurements, the trap frequencies were $(\omega_x, \omega_y, \omega_z) = 2\pi \times (67, 99, 244)$ Hz. For the evaporation, however, we started with near-degenerate molecules at $(\omega_x, \omega_y, \omega_z) = 2\pi \times (45, 67, 157)$ Hz and ended up, for example, at $(\omega_x, \omega_y, \omega_z) = 2\pi \times (52, 72, 157)$ Hz in case I or at $(\omega_x, \omega_y, \omega_z) = 2\pi \times (42, 56, 99)$ Hz in case II and case III (Fig. 3).

To measure the cross-dimensional thermalization, we heated the weakly bound molecules along the vertical direction after we separated them from unbound atoms and before STIRAP was applied. For this purpose, we used parametric heating by modulating the intensity of the 1,064-nm beam at twice the vertical trap frequency.

### Microwave-field generation

It is essential that the phase noise of the microwave source does not induce transitions between the dressed states. We generated the microwave with a vector signal generator (Keysight E8267D). The microwave passes through a voltage-controlled attenuator (General Microwave D1954) before it is amplified with a 10-W power amplifier (KUHNE electronic KU PA 510590 – 10 A). At 10-MHz carrier offset, we measured −150 dBc Hz$^{-1}$ phase-noise density from the signal generator and no significant enhancement from the amplifier. The microwave is emitted by a five-turn helical antenna (customized by Causemann Flugmodellbau) whose top end is about 2.2 cm away from the molecular sample.

With the voltage-controlled attenuator, we can adiabatically prepare the molecules in the dressed state by ramping the power attenuation linearly within 100 μs over a range of 65 dB.

### Imaging and thermometry

To image the molecules, we transferred them back into the non-dressed absolute ground state by ramping down the microwave power. Subsequently, the dipole traps were turned off and return STIRAP pulses were applied to bring the molecules back into the weakly bound state. After time of flight, typically 10 ms, the atoms were dissociated by ramping the magnetic field back over the Feshbach resonance. The magnetic field has to cross the Feshbach resonance slowly to minimize the release energy. In the end, the dissociated molecules were imaged by absorption imaging. We estimated that the derived temperature of the molecular sample could be overestimated by about 7 nK owing to the residual release energy. It is noted that the values of $T$ and $T/T_F$ reported in the main text do not account for the release energy.

To obtain the temperature of the molecular sample, we fit the absorption images with the Fermi–Dirac distribution

$$n_{FD}(x, z) = n_{FD,0} \; Li_2\left[-\zeta \exp\left(-\frac{x^2}{2\sigma_x^2} - \frac{z^2}{2\sigma_z^2}\right)\right], \tag{3}$$

where $n_{FD,0}$ is the peak density, $Li_2(x)$ is the dilogarithmic function, $\zeta$ is the fugacity and $\sigma_{i=x,z}$ are the cloud widths in the $x$ and $z$ directions. Given a cloud width $\sigma_i$, we can calculate the temperature $T_i$ by

$$\sigma_i = \frac{\sqrt{1 + \omega_i^2 t_{TOF}^2}}{\omega_i} \sqrt{\frac{k_B T_i}{m}}, \tag{4}$$

where $\omega_i$ is the trapping frequency in the $i$th direction, $t_{TOF}$ is the time of flight and $m$ is the mass of the molecules. The fugacity can be associated with the ratio of the temperature $T$ and the Fermi temperature $T_F$ with the relation

$$\left(\frac{T}{T_F}\right)^3 = -\frac{1}{6 \, Li_3(-\zeta)}, \tag{5}$$

where $Li_3(x)$ is the trilogarithmic function. $T_F = (6N)^{1/3} \hbar \overline{\omega} / k_B$ is given by the molecule number $N$ and the geometric mean trap frequency $\overline{\omega} = (\omega_x \omega_y \omega_z)^{1/3}$. By rewriting $\zeta$ and fixing $T_F$, we are left with only the fitting parameters $n_{FD,0}$, $T_x$ and $T_z$. We note that the temperature in the direction of the imaging beam $T_y$ is assumed to be equal to $T_x = T_h$.

In addition, we independently determine the temperatures of the molecular samples from the time-of-flight images by fitting the thermal wings of the cloud to a Gaussian distribution

$$n_{th}(x, z) = n_{th,0} \exp\left(-\frac{x^2}{2\sigma_x^2} - \frac{z^2}{2\sigma_z^2}\right), \tag{6}$$

where $n_{th,0}$ is the peak density. Similar to ref. [33], we first fit a Gaussian distribution to the whole cloud. We then constrain the Gaussian distribution to the thermal wings of the cloud by excluding a region of $1.5\sigma$ around the centre of the image. We find that by excluding $1.5\sigma$, the ratio of signal to noise allows for the fit to converge for all datasets in Fig. 3a.

The temperatures extracted from fitting the Fermi–Dirac distribution and fitting the Gaussian distribution to the thermal wings are compared in Extended Data Fig. 1.

### Model for elastic and inelastic collisions

The elastic and inelastic collision rate coefficients $\beta_{el}$ and $\beta_{in}$ are experimentally determined from the time evolution of the measured molecule number $N$, the average temperature $(2T_h + T_v)/3$ and the differential temperature $T_v - T_h$ by numerically solving the differential equations[19,34]

$$\frac{dN}{dt} = \left(-K \frac{2T_h + T_v}{3} n - \Gamma_1\right) N, \tag{7}$$

$$\frac{dT_h}{dt} = \frac{1}{12} K T_v T_h n + \frac{\Gamma_{th}}{3}(T_v - T_h), \tag{8}$$

$$\frac{dT_v}{dt} = \frac{1}{12} K (2T_h - T_v) T_v n - 2\frac{\Gamma_{th}}{3}(T_v - T_h), \tag{9}$$

with the mean density

$$n = \frac{N}{8\sqrt{\pi^3 k_B^3 T_h^2 T_v / m^3 \overline{\omega}^6}}. \tag{10}$$

Here, $K$ is the temperature-independent two-body loss coefficient, averaged for simplicity over all collision angles, and

$$\Gamma_{th} = \frac{n\sigma_{el}\upsilon}{N_{col}} \tag{11}$$

is the rethermalization rate with the elastic scattering cross-section $\sigma_{el}$ and the thermally averaged collision velocity

$$\upsilon = \sqrt{16 k_B (2T_h + T_v)/(3\pi m)}. \tag{12}$$

The average number of elastic collisions per rethermalization is taken from ref. [48] as

$$N_{col} = \overline{\mathcal{N}_z}(\phi) = \frac{112}{45 + 4\cos(2\phi) - 17\cos(4\phi)} \tag{13}$$

where $\phi$ is the tilt of the dipoles in the trap, which, in our case, corresponds to the tilt of the microwave wave vector with respect to the d.c. magnetic field. Following our characterization of the microwave polarization, we assume $\overline{\mathcal{N}_z}(29°) = 2.05$.

The anti-evaporation terms, that is, the first terms in equations (8) and (9), assume a linear scaling of the two-body loss rate with temperature. Our calculations predict that this assumption does not hold for small detunings ($\Delta < 2\pi \times 20$ MHz), as illustrated in Fig. 2. Our results, however, do not significantly change when we instead assume no temperature dependence in this regime.

Finally, after determining $\sigma_{el}$ and $K$, the elastic and inelastic collision rate coefficients

$$\beta_{el} = \sigma_{el}\nu \tag{14}$$

and

$$\beta_{in} = K(2T_h + T_v)/3 \tag{15}$$

are plotted in Fig. 2 assuming a fixed temperature $T = T_h = T_v$.

Example data of the loss measurements, performed to determine $\beta_{in}$, are shown in Extended Data Fig. 2a. At high densities, two-body loss is the dominant contribution, whereas at low densities, the exponential shape of the loss curve shows that one-body effects outweigh inelastic collisions. To limit the number of free-fit parameters, we determine $\Gamma_1 = 1.7(4)$ Hz in independent measurements at low densities, as shown in Extended Data Fig. 2b. To suppress confounding effects from inelastic collisions, we reduce the initial molecule number to about 2,000 for these measurements. Under these conditions, the 1/e lifetime is 570(100) ms without shielding, which is still mostly limited by residual two-body collisions. Turning on the shielding results in a similar 1/e lifetime of about 590(100) ms. The lifetime reduces to 300(50) ms when a microwave source with a 3 dB higher phase-noise density (Rohde & Schwarz SMF100A) is used. If we isolate the molecules by loading them into a three-dimensional optical lattice, we measure a lifetime of 8.0(1.2) s in absence of a microwave field, as shown in Extended Data Fig. 2c. Turning on the microwave field (using the Rohde & Schwarz SMF100A) results in a fast exponential decay to about half of the initially detected molecules, followed by a slow exponential decay. Assuming that the particles are isolated on individual lattice sites and that the faster decay is a result of mixing of two dressed states by phase noise of the microwave, we fit the data with the function

$$N(t) = N_0 e^{-t/\tau_0}\left(\frac{1}{2} + \frac{1}{2}e^{-t/\tau_{MW}}\right), \tag{16}$$

where $N_0$ is the initial number of molecules. We find a one-body loss time $\tau_0 = 4.4(1.4)$ s and a state-mixing time $\tau_{MW} = 210(90)$ ms, which is in reasonable agreement with the lifetime measurements in the bulk. The slightly faster decay might be caused by molecules in higher bands, leading to residual collisions in the lattice, which are not accounted for in equation (16). The scaling of the lifetime with the microwave phase noise in the bulk and the measurements in the lattice indicate that $\Gamma_1$ is currently limited by the noise power spectral density in the dressed-state transition (that is, at around $2\pi \times 10$ MHz offset from the carrier), even at a level of $-150$ dBc Hz$^{-1}$ (ref. [38]). In addition, we find that the 1,550-nm light, which is used to trap the molecules in the bulk (Fig. 1a), contributes with about 0.5 Hz to the one-body decay. The underlying loss mechanism is under investigation.

## Data availability
The experimental data that support the findings of this study are available from the corresponding author upon reasonable request. Source data are provided with this paper.

## Code availability
All relevant codes are available from the corresponding author upon reasonable request.

48. Wang, R. R. W. & Bohn, J. L. Anisotropic thermalization of dilute dipolar gases. *Phys. Rev. A* **103**, 063320 (2021).

**Acknowledgements** We thank Y. Bao, L. Anderegg, A. Pelster, A. Balaž and T. Shi for discussions; B. Braumandl for the simulation of the microwave field; F. Deppe and B. Wang for lending the ultralow-noise microwave signal generators; and T. Hilker for reading of the manuscript. We acknowledge support from the Max Planck Society, the European Union (PASQuanS grant number 817482) and the Deutsche Forschungsgemeinschaft under Germany's Excellence Strategy – EXC-2111 – 390814868 and under grant number FOR 2247. A.S. acknowledges funding from the Max Planck Harvard Research Center for Quantum Optics.

**Author contributions** All authors contributed substantially to the work presented in this manuscript. A.S., R.B. and X.-Y.C. carried out the experiments and improved the experimental set-up. A.S., R.B. and M.D. analysed the data. T.K. performed the theoretical calculations. I.B. and X.-Y.L. supervised the study. All authors worked on the interpretation of the data and contributed to the final manuscript.

**Funding** Open access funding provided by Max Planck Society.

**Competing interests** The authors declare no competing interests.

**Additional information**
**Correspondence and requests for materials** should be addressed to Xin-Yu Luo.

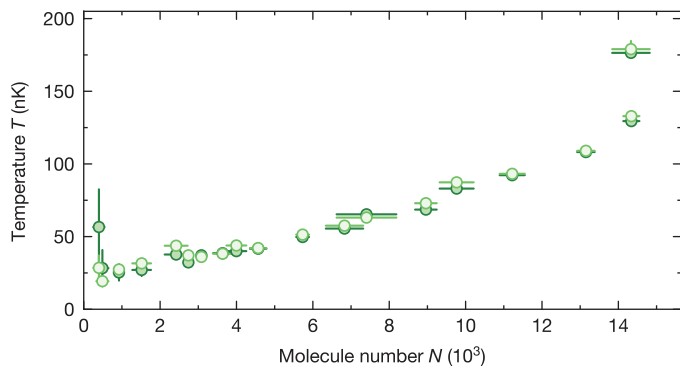

**Extended Data Fig. 1 | Thermometry.** Temperatures extracted from a
Fermi–Dirac distribution (dark) and a Gaussian distribution to the thermal
wings (bright) against the number of molecules $N$ after 150 ms evaporative
cooling for various final trap depths. The temperatures are extracted from
averaged images. The error bars of $N$ are the standard error of the mean of
5–20 repetitions. The error bars in $T$ are the standard deviation from the fit.

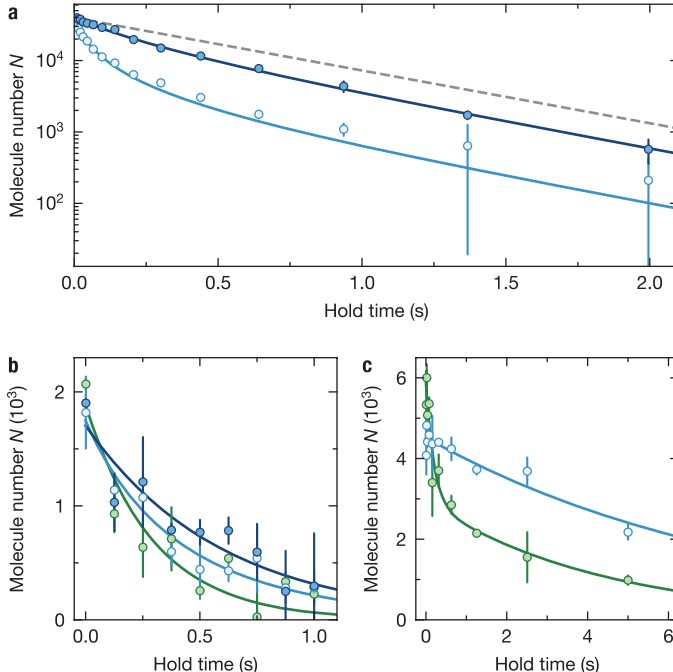

**Extended Data Fig. 2 | One- and two-body loss.** The bright blue data are taken without a microwave field, while the dark blue and dark green data are taken at Δ = 2π × 8 MHz using a Keysight E8267D and a Rohde & Schwarz SMF100A, respectively. **a**, Molecule loss at high initial densities. The grey dashed line shows the one-body contribution. The lines are fits to the differential-equation model. **b**, Loss at low initial densities. The one-body loss rate $\Gamma_1$ is determined from the measurement with shielding using the Keysight E8267D (dark blue). The lines are exponential fit functions. **c**, Loss in a 3D optical lattice. The data without (with) microwave are fitted using an exponential (a double-exponential) fit function. The error bars are the standard error of the mean of two (**a**) or three (**b**) repetitions.