## [Peer Review File · Nature]

Manuscript Title: Evaporation of microwave-shielded polar molecules to quantum degeneracy

Reviewer Comments & Author Rebuttals

Reviewer Reports on the Initial Version:

Referees' comments:

Referee #1 (Remarks to the Author):

This manuscript reports an exciting experimental advance in the field of ultracold molecules. The authors employed a blue-detuned microwave field to create repulsive barrier that successfully suppressed inelastic collision losses. The same microwave field induced strong, elastic dipolar interactions to enable the use of evaporative cooling to reach quantum degeneracy for a three-dimensional gas of NaK polar molecules. Particularly noteworthy is the achieved ratio of 460 for the elastic over inelastic collision rates.

This is only the second time that dipolar interaction assisted evaporative cooling of polar molecules have been realized in a three-dimensional geometry. In comparison to the previous work of KRb [Ref.32], which used a DC electric field-induced collision resonance for shielding the molecular loss, the NaK system demonstrates a significant advantage arising from its larger intrinsic dipolar moment. The demonstrated ratio of 460 for good over bad collisions is 40 times larger than what was achieved in Ref. 32. This favorable result helped improving the evaporation efficiency to reach degeneracy in 3D.

I congratulate the authors for this excellent achievement and I recommend publication of this paper in Nature. The experimental work reported here is of high quality, and the results are convincingly clear. This system reported here also represents the second quantum gas of ground state polar molecules. As already evidenced in the evaporation work, the large dipole moment of NaK, in comparison to KRb, make this an exciting new dipolar quantum gas system. I fully agree with the authors that this work joins other recent exciting advances in the field to signal the readiness of cold molecule systems for the exploration of a range of novel quantum phenomena and quantum information science.

Before the paper is accepted for publication, however, I have a few clarifications and suggestions for the authors to consider.

First, I find the discussion of one-body loss confusing. On page 2 of the main text, it is stated that "leading to a long life-time of the degenerate molecular sample of up to 0.6 s, mainly limited by residual one-body loss induced by the technical noise of the microwave." Well, first of all, I wouldn't call 0.6 s a long lifetime. Second, the origin of this lifetime limitation is apparently not due to the technical noise of the microwave. Quoting the results stated on the last page of Methods, "Under these conditions, the $1/e$

lifetime is 570(100) ms without shielding. Turning on the shielding results in a similar $1/e$ lifetime of about 649 (100) ms.” What limits the lifetime of 570(100) ms without the microwave field? The authors did not explain. Was this really a one-body loss? If so, by the vacuum in the chamber? Turning on the shield field resulted in a similar lifetime of 649(100) ms, essentially the same, statistically speaking. I hence do not understand how the authors came to the conclusion that their molecule gas life time is “mainly limited by residual one-body loss induced by the technical noise of the microwave”?

A similar statement of the lifetime is repeated on page 4, left column.

Page 5, Discussions. “Our numerical models, which agree well with the data outside of the hydrodynamic regime, predict a ratio of $\gamma \geq 1000$. This is almost a factor of 100 higher than the ratio realized in previous experiments in 3D [32] and almost an order of magnitude higher compared to experiments in 2D [31].” This comparison should be done by using the experimental realized value in this work vs “the ratio realized in previous experiments”. Otherwise the comparison is misleading. The direct experiment to experiment comparison would give a factor of 40 relative to the previous result in 3D reported in Ref. [32].

Overall the manuscript has presented a comprehensive list of citations of relevant work. I would however suggest that the authors rework the abstract and the introduction significantly. The main scientific achievement is the demonstration of microwave shielding to suppress the inelastic loss while enhancing the elastic dipolar interaction. From this perspective, the relevant state-of-the-art, over which the current work should be compared against, is the dipolar evaporation in 2D and 3D of KRb (ref. 31, 30, 32). In terms of quantum degeneracy, the relevant prior art is Ref. 22 and 23 (which is another excellent piece of work from the same group here). Association of deeply degenerate atomic gases has led to quantum degeneracy, thanks to the mediation of atom-molecule interactions during the association process. The statement of “While non-interacting polar molecules can partially inherit low entropy from degenerate atomic mixtures [21– 23], active cooling to the quantum degenerate regime has remained challenging” is thus unnecessarily confusing. Quantum degeneracy in 3D has been achieved in the prior work from both groups, period. What the authors have shown here is that they can turn on a strong tunable interaction in a 3D gas, a lot stronger than what’s demonstrated in Ref. [32]. I believe this is a sufficiently strong justification for publication in Nature.

Similarly, the statement in the abstract of “has so far prevented the cooling to quantum degeneracy in three dimensions” is also not fully appropriate. While it is true that “cooling to quantum degeneracy in 3D” had not been realized, if the authors wish to present an accurate picture of the state of the art, then they should simply state the fact that association to quantum degeneration has been achieved in 3D, cooling to degeneracy in 2D has also been achieved, and what they are demonstrating now is direct dipolar evaporation in 3D to reach quantum degeneracy. It’s a remarkable achievement, and they can help a reader right away by stating clearly what is new about this work at the outset. Finally, the statement of “Such unprecedentedly cold and dense samples of polar molecules” is also not fully accurate. A similar Fermi degeneracy value in 3D was already reported in Ref. 22.

Referee #2 (Remarks to the Author):

In this paper, the authors demonstrated microwave shielding in 3D NaK molecules. They used this shielding technique to evaporatively cool NaK molecules and achieved a quantum degenerate gas with $T/TF=0.36$. Using microwave shielding to suppress losses to perform evaporative cooling in 3D molecular gases is very interesting. The suppression of the inelastic losses with a ratio of 460 between elastic to inelastic collisions is impressive. The manuscript is publishable. However, for the following reasons, I am not fully convinced that the manuscript is suitable for Nature, at least in its present form.

1. The microwave shielding technique in 3D has been demonstrated in Ref. [35] using CaF molecules. The key technique is to use a high-power circularly polarized microwave wave field to dress the molecules, so that a blue-detuned barrier can be created. The idea and technique in the current manuscript are quite similar to the work in Ref. [35]. In this sense, it seems that the current work lacks originality.

2. For evaporative cooling, the authors mentioned that the starting point is a low-entropy gas and referred to their previous work (Ref. [23]). In Ref. [23], the authors claimed they prepared quantum degenerate gases of Feshbach molecules with $T/TF=0.28$, and low-entropy ground state molecules with $T/TF=0.52$. In the current work, after the evaporative cooling, they finally reached a temperature of $T/TF=0.36$. In this sense, they did not demonstrate "Evaporation of polar molecules to quantum degeneracy", as described in the title and abstract.

In the work from Jun Ye's group (Ref. [31], and [32]), they explicitly stated that the starting point of evaporative cooling is well above the Fermi temperature. In Ref. [31], they started from $T/TF=1.5$ and cooled the molecules to 0.6 TF. In Ref. [32], they started from $T/TF=2.0$ and ended with $T/TF=1.4$ TF.

In my opinion, the authors should also start from a temperature well above TF to demonstrate evaporative cooling to quantum degeneracy, as demonstrated in Ref. [31].

3. There are some inappropriate overstatements in the current manuscript. One example is the statement "corresponding to 0.36 times the Fermi temperature. Such unprecedentedly cold and dense samples" in the abstract. It is well known that in Ref. [22] quantum degenerate polar molecular gases with $T/TF=0.31$ have been demonstrated. The authors should be cautious when presenting their results.

Referee #3 (Remarks to the Author):

In this manuscript, the authors presented the very exciting results on direct and efficient evaporative cooling and the achievement of a quantum degenerate Fermis gas of NaK molecules. Although the

potential of ultracold polar molecules has been well recognized for some years, the experimental exploration has been lagging way behind due to the difficulties in creating them, and subsequently understanding and control their lossy collisions. In this work, the authors successfully implemented the loss control with blue detuned microwave coupling to the rotational levels which also induces efficient elastic dipolar collisions. The result of this is a high elastic to inelastic collision ratio which is vital for cooling the already ultracold 3D bulk sample of NaK down to the quantum degenerate regime.

While this is not the first experimental work on the microwave loss suppression of polar molecules, its successful demonstration to the 3D sample points to a general way for solving the loss problem. Other loss suppression methods, such as optical lattice isolation and electric field induced resonance, do exist, but the microwave method are more convenient and more compatible to typical ultracold apparatus. I thus have no doubt that this work should have a broad impact for the field as the loss problem is currently the main challenge many other cold molecule groups are facing.

Because of its importance, I believe this work can be published in a high impact journal such Nature. However, I also found that the manuscript itself can use some improvement before it can reach the standard of Nature. Some of the problems are:

1. In several places, the paper reads very redundant. For example, the first two paragraphs repeated many of the points already presented in the abstract.
2. In the abstract, the authors said, "Realizing their full potential requires cooling interacting molecular gases deeply into the quantum degenerate regime." Then the last sentence of the first paragraph "Many of these proposals require a deeply degenerate..." They are redundant and not fully consistent with each other.
3. Also in the abstract, why "test fundamental symmetries of nature" needs a degenerate sample?
4. The last paragraph of the discussion section starts with "The reduced temperature T/T_F reached here is only a factor of about four higher than the predicted critical temperature of topological p-wave superfluidity with strong dipolar interactions". After two sentences on how to improve the cooling further, the second half of the paragraph went on to discuss other possible things can be done with these further improvements.

These are just several examples. More concise and more logical discussions can be used in quite a few other places.

There are also several scientific questions:

1. With the detuned microwave coupling, the molecules could end up in a rotational mixture. What is the role of the rotational coherence here? Are there s-wave collisions between molecules in different rotational levels?
2. The issue of loss induced by technical issues is mentioned in several places but never made clear. A couple of sentences could be used to make the discussion more standalone and save the readers from looking it up in the supp. material.
3. It was noticed that ref. [23] is from the same group. It seems that a degenerate sample of NaK has been achieved there. Shouldn't the relation between the two works be described in a more

transparent way? By the way, the arxiv link for ref. [23] is pointing to a wrong paper.

4. In the first paragraph of the discussion, the comparison of gamma values between NaK and KRb is somewhat misleading. As these molecular species are very different from each other, the reason for the better performance in NaK is not because the current work did better or so on. So, instead of listing the numbers, it makes more sense to give the reason behind the difference.

Author Rebuttals to Initial Comments:

Response to the referee reports

First of all, we thank the referees for reviewing our work and their valuable comments and suggestions along with their constructive criticism, which helped us to further improve our manuscript. We are delighted to read that the referees found our results “very exciting” and “impressive”. We have revised the manuscript according to the referees’ suggestions. In the following response, we address the concerns raised by the referees point-by-point and explain the modifications we have undertaken as a result.

Note that the references changed with the resubmission. To be consistent with the original comments from the referees, we use here the citation numbers from the first submission.

Here we list the main revisions to the manuscript which are described in detail in the following responses:

1. We added 4 new references to the introduction. They include an early proposal on p -wave superfluidity with ultracold atoms, a proposal on optical shielding of molecule losses, a proposal on microwave shielding of molecule losses, and a study of thermalization and degeneracy of KRb molecules. We removed Ref. [21] during the revision.
2. We added subfigure c to Extended Data Fig. 2 showing lifetime measurements in a 3D optical lattice.
3. We added a new plot to Extended Data Fig. 2b to show the lifetime of the molecules with shielding using a Rohde & Schwarz SMF100A as signal generator.
4. We replotted the absorption images in the insets of Fig. 3d, e, and f. Initially we did not realize that the python function used to plot these images applies by default a filter that made the images appear somewhat fuzzy. We now deactivate this filter. Note this does not change the analysis of the images or any conclusion in the manuscript.
5. In the discussion section, we added a few sentences to clarify the confusion when comparing quantum degeneracy of ground-state molecules before and after turning on the interactions.
6. In the section about microwave shielding, we added a discussion to improve the explanation of the one-body loss mechanism.
7. We simplified the abstract and the first two paragraphs in the introduction.
8. We mirrored the image of the antenna and the laser beams in Fig. 1a as both, the helical antenna and the coordinate system, had the wrong handedness.

RESPONSE TO REPORT FROM REFEREE #1

First, I find the discussion of one-body loss confusing. On page 2 of the main text, it is stated that “leading to a long life-time of the degenerate molecular sample of up to 0.6 s, mainly limited by residual one-body loss induced by the technical noise of the microwave.” Well, first of all, I wouldn’t call 0.6 s a long lifetime.

(A1) We thank the referee for the criticism. Accordingly, we removed the word “long” from “long life-time”.

Second, the origin of this lifetime limitation is apparently not due to the technical noise of the microwave. Quoting the results stated on the last page of Methods, “Under these conditions, the $1/e$ lifetime is $570(100)$ ms without shielding. Turning on the shielding results in a similar $1/e$ lifetime of about $649(100)$ ms.” What limits the lifetime of $570(100)$ ms without the microwave field? The authors did not explain. Was this really a one-body loss? If so, by the vacuum in the chamber? Turning on the shield field resulted in a similar lifetime of $649(100)$ ms, essentially the same, statistically speaking. I hence do not understand how the authors came to the conclusion that their molecule gas life

time is “mainly limited by residual one-body loss induced by the technical noise of the microwave” ? A similar statement of the lifetime is repeated on page 4, left column.

(A2) This is a very good question. The $1/e$ lifetime of 570(100) ms without shielding can mostly be attributed to remaining two-body collisions. With microwave shielding the measured loss at low densities is dominated by the loss caused by the phase noise of the microwave. We performed additional measurements to further investigate this exponential loss mechanisms. Therefore, we have added a lifetime measurement of the dressed molecules in a deep 3D optical lattice as shown in a new subfigure of Extended Data Fig. 2. In absence of a microwave field, molecules frozen in the lattice exhibit a slow decay with a decay time of $\tau = 8$ s. In presence of the microwave field, we instead find a faster decay in a few hundreds of milliseconds followed by a slow decay. This indicates that the quality of our microwave source is — at present — a major limiting factor other than vacuum lifetime. In the future, we indeed anticipate further improvements in the single molecule lifetime with improved microwave source characteristics. We recently also found that our 1550-nm light source contributes to the one-body loss. The rate of this contribution is however only about 0.5 Hz (about 30% of the total one-body loss rate) at typical intensities used for the measurements in the paper. The reason for this loss contribution (laser noise, pump light from the fiber amplifier, etc.) is still under investigation. The 785.5-nm light lies between transitions to two vibrational states of the $A^1\Sigma^+$ potential. From a calibration measurement we deduced a one-body loss rate of at most 0.1 Hz for the largest intensity used for experiments presented in the paper. Following the suggestion by the referee, we extended the discussion about the one-body loss in the methods section and adapted the discussion accordingly in the main text.

Page 5, Discussions. “Our numerical models, which agree well with the data outside of the hydrodynamic regime, predict a ratio of $\gamma \geq 1000$. This is almost a factor of 100 higher than the ratio realized in previous experiments in 3D [32] and almost an order of magnitude higher compared to experiments in 2D [31].” This comparison should be done by using the experimental realized value in this work vs “the ratio realized in previous experiments”. Otherwise the comparison is misleading. The direct experiment to experiment comparison would give a factor of 40 relative to the previous result in 3D reported in Ref. [32].

(A3) Following the suggestion of the referee, we now compare the experimentally measured values of γ . The new text reads: “Our highest measured value of γ of about 500 is a factor of 40 larger than the ratio realized in previous experiments in 3D [32] and almost twice as large compared to experiments in 2D [31].”

Overall the manuscript has presented a comprehensive list of citations of relevant work. I would however suggest that the authors rework the abstract and the introduction significantly. The main scientific achievement is the demonstration of microwave shielding to suppress the inelastic loss while enhancing the elastic dipolar interaction. From this perspective, the relevant state-of-the-art, over which the current work should be compared against, is the dipolar evaporation in 2D and 3D of KRb (ref. 31, 30, 32). In terms of quantum degeneracy, the relevant prior art is Ref. 22 and 23 (which is another excellent piece of work from the same group here). Association of deeply degenerate atomic gases has led to quantum degeneracy, thanks to the mediation of atom-molecule interactions during the association process. The statement of “While non-interacting polar molecules can partially inherit low entropy from degenerate atomic mixtures [21– 23], active cooling to the quantum degenerate regime has remained challenging” is thus unnecessarily confusing. Quantum degeneracy in 3D has been achieved in the prior work from both groups, period. What the authors have shown here is that they can turn on a strong tunable interaction in a 3D gas, a lot stronger than what’s demonstrated in Ref. [32]. I believe this is a sufficiently strong justification for publication in Nature.

(A4) We thank the referee for the remark. Following the suggestion of the referee, we changed the wording to “While quantum degenerate gases of non-interacting molecules have been produced by assembling degenerate atomic mixtures [21– 23], direct cooling of interacting polar molecules to the quantum degenerate regime has remained challenging”. In the process, we replaced the citation of [Moses et al., Science 350, 659 (2015)] with [Tobias et al., Phys. Rev. Lett. 124, 033401 (2020)].

Similarly, the statement in the abstract of “has so far prevented the cooling to quantum degeneracy in three dimensions” is also not fully appropriate. While it is true that “cooling to quantum degeneracy in 3D” had not been realized, if the authors wish to present an accurate picture of the state of the art, then they should simply state the fact that association to quantum degeneration has been achieved in 3D, cooling to degeneracy in 2D has also been achieved, and what they are demonstrating now is direct dipolar evaporation in 3D to reach quantum degeneracy. It’s a remarkable achievement, and they can help a reader right away by stating clearly what is new about this work at the outset. Finally, the statement of “Such unprecedentedly cold and dense samples of polar molecules” is also not fully accurate. A similar Fermi degeneracy value in 3D was already reported in Ref. 22.

(A5) Following the suggestion of the referee, we clarified the quoted wording to “has so far prevented direct cooling via elastic collisions to quantum degeneracy in three dimensions”. As we mentioned in the previous reply, we rewrote the first sentence of the second paragraph to make “association to quantum degeneration has been achieved in 3D” more transparent to readers. “cooling to degeneracy in 2D has also been achieved” is indeed mentioned in the third paragraph. We decided not to repeat these two statements in the abstract to keep the abstract length reasonable. Following the suggestion by referee 1 as well as referee 3, we removed the word “unprecedentedly” from the last sentence of the abstract.

RESPONSE TO REPORT FROM REFEREE #2

1. The microwave shielding technique in 3D has been demonstrated in Ref. [35] using CaF molecules. The key technique is to use a high-power circularly polarized microwave wave field to dress the molecules, so that a blue-detuned barrier can be created. The idea and technique in the current manuscript are quite similar to the work in Ref. [35]. In this sense, it seems that the current work lacks originality.

(B1) It is correct that shielding via circular polarized microwave was previously demonstrated in Ref. [35]. However, while extending the lifetime of a pair of molecules in comparison to the lifetime without microwave field (i.e., shielding) is very much desirable, we think an equal or even bigger impact on our field is given by the possibility to realize strong dipolar interactions and simultaneously achieve a reasonably long lifetime of the molecular sample, e.g., for evaporative cooling. As one can see, e.g., from Fig. 3 of Ref. [17], realizing dipolar interactions in a conventional way (i.e., using a near-resonant microwave field or a DC electric field in 3D) is accompanied by a strong increase in the inelastic collision rate, so that there are at best a few elastic collisions per inelastic collision. While Reference [35] demonstrated a suppression of losses, it only *calculated* the elastic scattering rate but did not demonstrate it experimentally. In our experiment, on the other hand, we measured the elastic collision rate, showed the tunability of the dipolar interaction and applied the interactions in order to realize evaporative cooling. We therefore think that our work very much complements the work presented in Ref. [35] and goes well beyond it in several important points.

2. For evaporative cooling, the authors mentioned that the starting point is a low-entropy gas and referred to their previous work (Ref. [23]). In Ref. [23], the authors claimed they prepared quantum degenerate gases of Feshbach molecules with $T/T_F=0.28$, and low-entropy ground state molecules with $T/T_F=0.52$. In the current work, after the evaporative cooling, they finally reached a temperature of $T/T_F=0.36$. In this sense, they did not demonstrate “Evaporation of polar molecules to quantum degeneracy”, as described in the title and abstract.

(B2) We thank the referee for the criticism. Based on the comments by the referee, we believe it is indeed helpful to expand a bit more on the initial conditions for evaporative cooling achieved in experiments so far. In general, ground-state molecules do not thermalize without (dipolar) interactions. As correctly pointed out by the referee, the value of $T/T_F = 0.28$ from Ref. [23] is the degeneracy of the Feshbach molecules right after their formation. It is also the apparent degeneracy of the ground-state molecules measured by analyzing the momentum distribution right after STIRAP that was inherited from the Feshbach molecules. However, the STIRAP process of finite fidelity, invariable produces a non-thermalized sample (which can be very close to a thermalized one depending on the STIRAP efficiency). This finite STIRAP efficiency can be modeled as a particle loss in the many-body system, which, however, for non-interacting particles does not change the momentum distribution. Thus for ground-state molecules, the value of $T/T_F = 0.52$ is a more realistic estimate of the degeneracy, including effects from 40% loss of particles during the preparation of the pure sample of Feshbach molecules and during STIRAP (similar to the correction presented by the JILA team in the supplementary material of [PRL 124, 033401 (2020)] for KRb).

In addition, when establishing stable thermalizing interactions via microwave shielding, we noticed during preparation that the molecular sample is also excited in other ways that do not directly show up in the momentum distribution of the non-interacting cloud. For example, a sloshing motion is induced by the photon-recoil momentum transfer from the STIRAP pulse. Also changes in the aspect ratio of the trap can effectively excite the system in absence of cross-dimensional thermalization. It is important to note that only after stable interactions have been established and after any residual excitations have been dampened out and the system has fully thermalized can a reliable experimental measurement of the degeneracy and temperature of the system via its momentum distribution be performed.

Furthermore, remaining inelastic collisions and one-body loss reduce the number of molecules from 2.5×10^4 to 1.4×10^4 during the rethermalization. If we don't force evaporation but just hold the molecules in the trap for 150 ms, the combination of all aforementioned effects leads to a thermal sample close to T_F , as shown in case I in Fig. 3. Although those heating problems can be minimized in the future, the evaporation is essential to cool the sample to

a deep degeneracy of $0.36 T_F$ in the present work. Our experiments therefore clearly demonstrates 3D evaporation of an interacting molecular sample from $T/T_F \approx 1$ to $T/T_F = 0.36$, with both degeneracy measurements based on *thermalized samples*.

As both referee 2 and 3 raised similar questions regarding the quantum degeneracy of the ground-state molecules, we accordingly added a few sentences in the discussion section to explain the relation between Ref. [23] and the present work. Besides that, in order to make the context in the section of evaporation more understandable, we replaced the phrase “This motion damps out while the molecules thermalize, which effectively reduces the phase space density of the sample.” with “Damping of such collective excitations and particle loss significantly reduce the phase space density of the sample.”

In the work from Jun Ye’s group (Ref. [31], and [32]), they explicitly stated that the starting point of evaporative cooling is well above the Fermi temperature. In Ref. [31], they started from $T/TF=1.5$ and cooled the molecules to $0.6 TF$. In Ref. [32], they started from $T/TF=2.0$ and ended with $T/TF=1.4 TF$. In my opinion, the authors should also start from a temperature well above TF to demonstrate evaporative cooling to quantum degeneracy, as demonstrated in Ref. [31].

(B3) The goal of the present work (and we believe the main goal of the field) is to prepare the most degenerate interacting samples of dipolar molecules. To achieve this, it would be counterproductive to artificially heat the sample prior to the evaporation. After all, the sample has a limited amount of molecules and if the number of molecules gets too low, the evaporation becomes inefficient (as shown in Fig. 3a). As noted above, our experiment clearly demonstrates 3D evaporation of an interacting molecular sample from $T/T_F \approx 1$ to $T/T_F = 0.36$, very much in line with the ranges the referee mentions, but to even lower temperatures.

Also, to our knowledge, the starting value of T/T_F in Ref. [32] was not artificially enhanced to improve the demonstration of the evaporative cooling, but preparing the interacting samples in these experiments came at the cost of an increased T/T_F . The authors in Ref. [32] wrote “Compared with the procedure in ref. 35, which produced a degenerate Fermi gas at $T/T_F = 0.3$ with $N_{\text{KRb}} = 2.5 \times 10^4$, the present approach requires preparing molecules in $|1, 0\rangle$ at E_S . This requires a ramp of the electric field that causes molecular loss and heating, limiting the highest PSD achieved in this work. Future technical improvements, such as direct creation of molecules at $|E_S|$, will enable evaporation of molecular gases to deep quantum degeneracy.” Both our work and Ref. [32] have to deal with heating and loss of molecules when turning on interactions although the underlying mechanisms differ to some degree. Again it shows that the measured quantum degeneracy of non-interacting samples is not necessarily maintained once interactions rethermalize the sample and evaporative cooling is essential to reach deep degeneracy.

3. There are some inappropriate overstatements in the current manuscript. One example is the statement “corresponding to 0.36 times the Fermi temperature. Such unprecedentedly cold and dense samples” in the abstract. It is well known that in Ref. [22] quantum degenerate polar molecular gases with $T/TF=0.31$ have been demonstrated. The authors should be cautious when presenting their results.

(B4) Note that the quoted value of $T/T_F = 0.31$ from Ref. [22] reflects the degeneracy of the Feshbach molecules in the association process (before the sample becomes non-interacting). An estimate of the degeneracy of the ground-state molecules for KRb that takes into account the finite efficiency of the STIRAP transfer is presented in the follow-up publication by the JILA team [Fig. 5 in the supplementary material of PRL 124, 033401 (2020)], which gives $T/T_F = 0.37$ assuming 90% STIRAP efficiency. In addition, with our recently gained experience on shielded molecules, we find that a fair comparison of degeneracy between samples of ground-state molecules can only be realized if both samples are indeed interacting and thermalized after the ground-state transfer (see above discussion). Otherwise, any residual excitations of the sample might be overlooked, as they do not necessarily manifest in the momentum distribution. However, since referee 1 and 3 raised similar concerns we decided to remove the word “unprecedentedly” from the text.

RESPONSE TO REPORT FROM REFEREE #3

1. In several places, the paper reads very redundant. For example, the first two paragraphs repeated many of the points already presented in the abstract.
2. In the abstract, the authors said, “Realizing their full potential requires cooling interacting molecular gases deeply into the quantum degenerate regime.” Then the last sentence of the first paragraph “Many of these proposals require a deeply degenerate. . .” They are redundant and not fully consistent with each other.

(C1) Following the suggestion of the referee, we have extensively rewritten the abstract and introduction to reduce repetition and to avoid confusion. “Many of these proposals require ...” is now specialized to the requirements for quantum simulation.

3. Also in the abstract, why “test fundamental symmetries of nature” needs a degenerate sample?

(C2) It is true that precision experiments do not necessarily always require degenerate samples. But cold samples with high phase-space density would e.g. allow to store the molecules in a weak optical lattice with high filling factor. In this way, long interrogation times with many molecules can be realized. Furthermore, confining the molecules in a small region of space will minimize detrimental spatial inhomogeneities and engineering entangled many-body states could lead to even higher precision in the future. Similarly, strontium optical clocks [Science 358, 90 (2017)] do not require degeneracy but certainly benefit from it and future exciting work towards clocks based on entangled many-body states will certainly require it. We thus believe the general perspective of this sentence, especially for future experiments, to be a valid statement.

4. The last paragraph of the discussion section starts with “The reduced temperature T/T_F reached here is only a factor of about four higher than the predicted critical temperature of topological p-wave superfluidity with strong dipolar interactions”. After two sentences on how to improve the cooling further, the second half of the paragraph went on to discuss other possible things can be done with these further improvements. These are just several examples. More concise and more logical discussions can be used in quite a few other places.

(C3) We removed the quoted sentence from the discussion as its content is essentially repeated later in the paragraph. We also went through the manuscript and improved the flow in various places. We thank the referee for their comment which helped us to improve the manuscript a lot.

There are also several scientific questions:

1. With the detuned microwave coupling, the molecules could end up in a rotational mixture. What is the role of the rotational coherence here? Are there s-wave collisions between molecules in different rotational levels?

(C4) The dressed states are a result of the microwave coupling between the rotational states, they indeed represent a time-dependent superposition of rotational states. We added the definition of the dressed states to the paper (Eq. 1) to clarify this point. The phase between the rotational state components in the dressed state is fixed by the microwave field. As long as the molecules stay in a pure dressed state, they remain indistinguishable fermions. If, however, microwave phase noise couples the dressed states, the molecules can become distinguishable and undergo an inelastic s-wave collision. This process is limited by the slow coupling between the dressed state and therefore manifests as a slow loss with one-body characteristic, i.e., exponential decay.

2. The issue of loss induced by technical issues is mentioned in several places but never made clear. A couple of sentences could be used to make the discussion more standalone and save the readers from looking it up in the supp. material.

(C5) Following the suggestion of the referee, we simplified the main text and added the following discussion of the one-body loss in the section of shielding “The latter is mainly caused by the coupling to other dressed states due to the phase noise of the microwave, which results in an exponential decay with a time constant of about 600 ms when the two-body loss is small (Methods).”

3. It was noticed that ref. [23] is from the same group. It seems that a degenerate sample of NaK has been achieved there. Shouldn't the relation between the two works be described in a more transparent way?

(C6) Following the suggestion of the referee and comment 2 from referee 2, we improved the discussion about the sample thermalization in the evaporation section and added a few sentences to the discussion that set the presented results in relation to the results reported in Ref. [23].

It is important to note that while Ref. [23] indeed produced a quantum-degenerate sample of ground-state molecules, the degeneracy of the sample cannot easily be further improved. While the loss of phase-space density between the association of Feshbach molecules and the start of the evaporation could be minimized to a certain level, evaporative cooling is essential to cool the molecules to deep degeneracy (see also above discussion in the reply to Referee 2).

By the way, the arxiv link for ref. [23] is pointing to a wrong paper.

(C7) We thank the referee for pointing out this embarrassing mistake. We have fixed the arXiv link accordingly.

4. In the first paragraph of the discussion, the comparison of gamma values between NaK and KRb is somewhat misleading. As these molecular species are very different from each other, the reason for the better performance in NaK is not because the current work did better or so on. So, instead of listing the numbers, it makes more sense to give the reason behind the difference

(C8) It is certainly not our intention to claim that our experiment was carried out more precisely than the experiments in Ref. [32] and [33]. With regard to a comment from referee 1, we rewrote this paragraph. The improvement of the gamma ratio mainly comes from the large effective dipole moment of the NaK molecules. Meanwhile the mechanisms suppressing the inelastic collisions are vastly different therefore it is not straightforward to give a direct comparison. Following the suggestion of the referee, we added a new sentence in the beginning of the discussion section “As a result of the large effective dipole moment of the microwave-dressed NaK molecules.” We hope that this will help the reader to understand the reason behind the difference in gamma ratios. We also removed the comparison of the temperatures which is not straightforward as the lowest temperatures after evaporation also depend on geometry and initial phase space densities.

Reviewer Reports on the First Revision:

Referees' comments:

Referee #1 (Remarks to the Author):

I am satisfied with the revision of the manuscript, particularly with a detailed clarification of the one-body loss mechanism together with additional data. I recommend publication of the manuscript in its present form.

Referee #2 (Remarks to the Author):

I find my concerns have been well addressed. I am happy to recommend the current manuscript for publication in Nature.

Referee #3 (Remarks to the Author):

The authors have addressed all my questions clearly. I believe this work is now ready to be accepted.